# Workplace Ostracism Effects on Employees’ Negative Health Outcomes: Focusing on the Mediating Role of Envy

**DOI:** 10.3390/bs13080669

**Published:** 2023-08-10

**Authors:** Hyunghoon Kim, Eunmi Jang

**Affiliations:** College of Business, Honam University, Gwangju 62399, Republic of Korea; khh@honam.ac.kr

**Keywords:** workplace ostracism (WO), envy, job stress, burnout

## Abstract

Employee health is crucial to organizational success. However, workplace ostracism (WO) has significant negative effects on employee health. Numerous researchers have extensively examined how WO influences employees’ negative health (job stress, burnout); however, the focus on mediating effects in the relationship between WO and health has been lacking. This study examined the cognitive evaluation response to WO by employees who perceive they have been ostracized because another employee envies them. The psychological defense mechanism is expected to be activated—thus triggering job stress and burnout. We investigated envy perceived by individuals as a mediator of WO, job stress, and burnout using data from a 2-wave longitudinal survey of 403 employees of a South Korean firm. We found that employees perceived WO. Specifically, based on the sensitivity to being the target of a threatening upward comparison theory, it was confirmed that envy was a mediator in the relationship between WO and negative health outcomes. Our results are the first to show that the perception of envy can mediate the maintenance of a positive self-image in the context of WO in South Korea. The results suggest that a greater awareness of and focus on envy, and WO is required.

## 1. Introduction

Employees are an important human resource in organizations. Their personal health is a fundamental issue at the organizational level. However, results from the scientific literature indicate that ostracization causes psychological pain in individuals and negatively affects their behavior and mental and physical health [1,2,3]. When high-performing workers like Ella experience psychological distress, such as stress or burnout, their self-esteem is lowered, and their health suffers [1,4]. This causes disruptions in their work progress and eventually results in additional costs for the organization. As such, it is important to focus on employees’ health in relation to workplace ostracism (hereafter, “WO”).

Recently, WO has been recognized as a severe organizational problem behavior [5]. Previous studies have found that a surprisingly large number of people are being ostracized within their organizations. For example, in a study on how being ostracized by co-workers affects people’s attitudes and behaviors at work, 13% of the survey respondents reported experiencing some neglect or rejection [6]. Another study by Fox and Stallworth [7], in which 66% of respondents reported experiencing a similar experience of bullying, suggested that bullying in organizations is currently a behavior that anyone can engage in. Workplace bullying has, therefore, become prevalent [8].

Why do problem behaviors, such as WO, become widespread in organizations? One reason is that ostracism is an unobtrusive behavior and, therefore, is not subject to punitive action [4]. Thus, WO can be problematic because it is likely to persist among coworkers for a long time. Nevertheless, WO as an organizational behavior remains relatively unexplored, and many unresolved research questions have been raised [5]. South Korea, a small but strong country that has rapidly become prosperous, has the stigma of having the highest suicide rate among the OECD countries [9]. As extreme incidents have recently occurred due to WO, its cause should be identified through further research; hence, it is timely to study the cause and mechanism of WO by targeting office workers in South Korea.

Our study aims to examine WO as a mechanism of job stress and burnout affecting employees’ health and suggest theoretical and practical implications for organizations. We argue that individual perceptions of WO may differ from person to person according to their perceptions of the source of ostracism. The experience of becoming aware of WO is psychologically very painful for the victim. WO is dangerous because it is not obvious. As an individual recognizes that they are experiencing WO, the psychological defense mechanism may be activated as a cognitive evaluation reaction, with the individual perceiving that the WO is occurring because someone within their organization is envious of them. This can lead to job stress and burnout. Our argument is in line with research by Balliet and Ferris [10]. As mentioned earlier, Ella believes she is being bullied because she perceives that she is an object of envy among her peers as she is a high performer. However, it is the position that there is no need to react negatively or repulsively to her perception itself [5]. Individuals activate cognitive thinking about themselves in threatening situations, such as social exclusion, in an attempt to determine the reasons behind the threat. Meaningful actions are then based on these reasons [11].

To support these arguments, we suggest a research model based on social comparison. When individuals face complex or ambiguous situations such as WO, they compare themselves with others to convince themselves of their reality [12]. For example, in Ella’s case, the situation in which she was being ostracized made her believe that her co-workers envied her work-related success and the approval of her superiors. This is because when humans encounter undesirable social information, such as ostracizing, they momentarily perceive it as a threat, and their self-protection becomes important [13,14,15]. This process is at the core of our study. Humans activate the “psychological immune system” when they infer or rationalize why someone is doing something to them in a threatening and uncomfortable situation [16]; that is, when there is a detrimental effect on an individual, they offset negative social experiences by focusing on success (e.g., [17,18]). Through this mechanism, employees who believe they are successful at work and contribute to the organization may perceive themselves as objects of envy by their peers [19,20].

To date, studies related to ostracism have mainly investigated job stress or burnout as parameters (e.g., [21,22,23,24]). We aim to go a step further and present a model that provides deeper insights by demonstrating the mediating effect of envy in the relationship between perceived ostracization and stress or burnout. This study aimed to demonstrate that WO positively (+) affects employees’ job stress and burnout and that an individual’s perception of themselves as an object of envy has a mediating effect between WO and employee health outcomes (job stress and burnout). This mediating effect is a unique perspective that has not yet been explored in WO research. We aim to make theoretical and practical contributions to the development of WO research by identifying gaps in the current literature regarding the mediating effect of the perception of envy.

## 2. Theoretical Background

When people are mistreated, they attempt to understand the reasons for this mistreatment [25]. Employees who are aware of ostracism in their organizations are no exception; they also look for a reason for being ostracized. As a result, among the various interactions that occur in interpersonal relationships, the interpretation of a discretionary phenomenon such as WO affects individual responses [26]: the victim of WO will likely determine a reason for being a target, and this recognition will affect their job stress and burnout as aspects of their personal health.

When an employee achieves exceptional job performance, they may become the object of envy in the workplace [27]. Interestingly, as humans are likely to falsely perceive that others envy them [28]; although this may not be the case, this perception can occur even if the employee is not necessarily a high-performing worker. Thus, this study focuses on individual perceptions that influence one’s reactions rather than the objective characteristics of the reality in the workplace. How an individual perceives reality or a phenomenon influences their subsequent behaviors and psychological health (cf. Kristof-Brown, Zimmerman, and Johnson’s [29] meta-analysis results [30,31]); therefore, we argue that when an individual who experiences WO perceived that they are the object of envy, the resulting emotional response harms their health [32,33,34].

### 2.1. Workplace Ostracism

WO has been studied under organizational behavior since the 1970s [35] and has been treated as an abnormal social phenomenon. Negative social phenomena similar to WO include organizational misbehavior [36], antisocial behavior [37], aggression [38], dysfunctional behavior [39], workplace deviance [40], counterproductive work behavior [41], social undermining [42], and workplace bullying [7,43]. Although ostracism has long been studied in various social science fields [44], the separate distinction of “in the workplace” is added in the context of organizational behavior to emphasize its importance. As a concept, it has been defined by Robinson, O’Reilly, and Wang [45] as “an individual or group within the workplace that decides that it is socially acceptable to do so, omitting the participation of a specific member”. One example is treating a particular member of the organization as invisible, which includes omitting ambiguous behaviors, such as not making eye contact with them or not inviting them to formal or informal gatherings within the team [3].

Although highly undesirable and a negative social phenomenon among employees, WO differs from other dysfunctional behaviors in organizations that include acts of exclusion, workplace harassment, incivility, bullying, deviance, and social undermining [45], which can largely be observed as interactions. In contrast, WO is characterized as being non-interactional and not blunt or overt; in other words, openly negative words or actions are not included in this behavior [45]. WO is more ambiguous than other dysfunctional behaviors [8]; hence, it is much more challenging to deal with and can be more threatening than incivility, harassment, or bullying. As such, O’Reilly et al. [46] suggest that ostracism is more harmful than other forms of workplace abuse. More research is needed on WO because of its possible potential adverse effects on employees; however, it has received relatively little attention to date [1].

### 2.2. Workplace Ostracism and Employees’ Negative Health Outcomes

Employees are key organizational resources, and job stress and burnout are representative negative health indicators that need to be managed. Derived from the Latin word “*stringere*”, meaning narrow or suppressed, the word stress was first used in the 14th century. Stress occurs in objective interactions but as a subjective experience [47]. It refers to the physical and psychological reactions that occur when pressure or threat is perceived. Among various stresses, job stress can be defined as “harmful physical and mental reactions that occur when job requirements do not match the ability, resources, or desires of workers” (International Institute for Occupational Safety and Health, NIOSH). Job stress can significantly impact individual health [48], with all job stress parameters explaining 41% of general health-related changes [49]. Burnout is a state of chronic stress that involves an individual becoming “exhausted due to excessive demands on energy, strength, or resources” at work [50]. It is defined as a psychological syndrome of emotional exhaustion, dehumanization, and reduced achievement [51]. Therefore, we assumed that two negative aspects related to health, job stress and burnout, are important variables and examine these with WO.

WO is an interpersonal stressor [8,48,52]. Both theoretical and empirical studies support its potential negative impact on individual health. According to the conservation of resources theory (COR), individuals have limited resources that they strive to possess, protect, and further establish [53,54]. However, individuals that are aware of WO not only have to mobilize their resources to fight back but also use them in isolation. Resource depletion occurs because the likelihood of replenishment is low; therefore, employees deprived of resources because of WO are more likely to become stressed and exhausted [53]. Social support frees employees from the harmful effects of stressful experiences. As WO indicates a lack of social support, individuals who experience ostracism are less likely to cope with stressful work experiences [55,56,57].

WO is a stressor that is also consistent with affective events theory, which states that employees react emotionally to specific work events and that such emotional reactions are important determinants of employee attitudes and behaviors [58,59]. Workplace exclusion is similar to WO as it can create negative emotional states that increase the experience of stress and reduce an individual’s resources [60]. This exacerbates the imbalance between job demands and resources and increases the likelihood of burnout. The most widely reported component of burnout is a chronic state of emotional and physical exhaustion [61], which includes the feeling that one’s emotional and physical resources are overextended and depleted [62]. Emotional burnout occurs when emotional demands exceed an individual’s ability to manage interpersonal relationships [63]. Workplace ostracism constitutes a loss of resources in terms of unsupportive colleagues. Research has shown that individuals experience emotional exhaustion when they do not have sufficient resources to handle daily tasks [64]. When employees are ostracized, they lose their emotional connections with others. Humans need social interaction to share emotional feelings, strengthen emotional resources, and maintain psychological and physical health [65]. Emotional resources are lost when the need for emotional sharing is unmet, which leads to emotional burnout. This suggests a unique relationship between WO and emotional exhaustion.

Several empirical studies have revealed an association between ostracism and health. Individuals who perceive WO have lower levels of satisfaction and psychological health [1,6]. According to Wu et al. [66], it is essential to investigate the association between WO and stress-related outcomes. Ostracism is related to experiencing a lack of control over a situation that leaves the victim feeling helpless in the workplace. Overall, ostracism is considered a workplace stressor that threatens the employee’s ability to cope with work and daily life. Based on the COR theory [54] and Williams’s ostracism model [52], the following hypotheses were established regarding the effects of WO:

**Hypothesis** **1.**
*WO is strongly related to employee’s negative health outcomes.*


**Hypothesis** **1(a).**
*WO is strongly related to employee’s job stress.*


**Hypothesis** **1(b).**
*WO is strongly related to employee’s burnout.*


### 2.3. Mediation Effect of Envy

When faced with the threatening situation of WO, a self-protective thought named ‘envy’ is often evoked as a coping mechanism. Envy is a unique emotion that strongly influences an individual’s behavior [67]. It cannot be seen as a simple personality trait but as a relative emotion caused by someone else [68]; therefore, an individual’s perception that they are the object of someone’s envy should also be considered a temporary and circumstantial phenomenon [67]. The individual who recognizes WO activates a perception of others’ envy as a self-protective thought, which may later lead to job stress or burnout.

To support our argument, we adopted the perspective of sensitivity to being the target of a threatening upward comparison theory in a broad framework. Upward comparison is the perception that someone has a more significant advantage than oneself [12]; it arises from a competitive paradigm with others and has negative effects on the self [69]. High-performing employees who recognize ostracism in their organizations may have self-protective thoughts that their co-workers envy their achievements and are jealous of their success. They view their image favorably because they need psychological defense mechanisms to buffer negative experiences related to ostracism [16]. Menon and Thompson [70] suggest that when individuals experience social exclusion similar to WO, they struggle to find a reason for it and attempt to make cognitive and emotional judgments about others’ motives for their exclusion. They subsequently want to maintain their positive emotions to respond to the threat [71], as WO is a painful experience, both psychologically and attitudinally, for the person concerned [71]. As such, a psychological immune system is necessary [44]. Several studies have shown that social exclusion similar to WO induces a response to defend oneself (e.g., [72,73,74]).

An individual’s perception that someone in their organization is envious of them can become a stressful experience that interferes with their work progress (e.g., [19,69,75]) and leads to burnout. Since the Republic of Korea is a society with a high tendency to be relationship-oriented, the negative influence of an individual’s perception of not having a good relationship with someone can be significant. Thus, an individual’s belief that they are envied by their co-workers can have a detrimental effect on their job stress and could lead to burnout. In summary, employees who believe they are experiencing WO will recognize job stress and burnout as negative influences. In this process, they will activate the perception that they are the object of envy through upward comparison as a form of psychological immunity to protect themselves. Based on the results of previous studies, the following hypotheses were established regarding the mediating effect of envy (see Figure 1):

**Hypothesis** **2.**
*The relationship between WO and an employee’s health outcomes is mediated by the perception that they are the target of co-worker envy.*


**Hypothesis** **2(a).**
*The relationship between WO and an employee’s job stress is mediated by the perception that they are the target of co-worker envy.*


**Hypothesis** **2(b).**
*The relationship between WO and an employee’s burnout is mediated by the perception that they are the target of co-worker envy.*


## 3. Methods

### 3.1. Participants and Procedures

We conducted an online survey with 530 employees who worked at various companies in South Korea to validate the hypotheses. The online survey was available for five weeks, providing ample opportunity for participants to complete it. The first and second rounds were conducted with a one-month time difference. The total number of participants was 520 in the first survey. These 520 participants were requested to participate in the second survey, and 430 responses were received. After excluding missing values, data from 403 participants were used for the final analysis (see Table 1).

To solve the problem of the common method bias [76], the survey was conducted twice with a time difference. The first survey measured workplace ostracism and envy, and the second measured job stress and burnout. By adopting this research design, we overcame the limitations of cross-sectional research.

### 3.2. Measures

The surveys employed a 5-point Likert scale. We translated the original English questionnaires into Korean. To ensure the reliability and validity of the research tool, we followed a standard translation and back-translation procedure [76].

***WO.*** We measured employees’ perceptions of WO using Ferris et al.’s [1] 10-item scale. A sample item is “Others at work treated you as if you were not there”.

***Envy.*** We measured employees’ perceptions of being envied using Vecchio’s [28] 3-item scale. A sample item is “Because of my success at work, I am sometimes resented by my coworkers”.

***Job stress.*** We measured the employees’ perceptions of job stress using Keller’s [77] four-item scale. A sample item is “I experience stress from my job”.

***Burnout.*** We measured employees’ perceptions of burnout using Kalliath et al.’s [78] five-item emotional exhaustion scale. A sample item is “I feel emotionally drained from my work”.

### 3.3. Statistical Analysis

First, we performed descriptive and correlation analyses to assess the data’s normality assumption, encompassing skewness and kurtosis statistics. No significant concerns arose during this stage. The data underwent a two-step procedure for structural equation modeling (SEM) [79]. Firstly, a confirmatory factor analysis (CFA) was executed to evaluate the reliability and validity of the measurement model. Subsequently, SEM was conducted to examine the connections between the study variables. For these analyses, we used the SPSS 25 and AMOS 25 software packages.

## 4. Results

### 4.1. Measurement Model Assessment

CFA assessed the measurement model’s psychometric properties. The results showed that the measurement model fitted the data well: χ^2^ = 577.09, χ^2^/*df* = 1.62, CFI = 0.94, TLI = 0.93, RMSEA = 0.07. The reliability of the measures was evaluated by examining Cronbach’s alpha coefficients and composite reliability (CR) values (see Table 2) [80]. The results showed that the scales exhibited good reliability as Cronbach’s alpha coefficients of all constructs were more than acceptable, ranging from 0.87–0.95 [31] and, therefore, meeting the threshold requirement (>0.70). Factor loadings and average variance extracted (AVE) values were employed to examine the construct and convergent validity of the measures. All factor loadings of the measures showed high significance, with the smallest factor loading being 0.66. The AVE values were greater than 0.50 (ranging from 0.59–0.82), satisfying the threshold criteria. In conclusion, the measurement model exhibited sufficient psychometric properties.

A Pearson correlation analysis was performed to examine the relationship between the study variables (Table 2). The findings indicated significant positive correlations between WO and three variables: envy (r = 0.33; *p* < 0.001), job stress (r = 0.17; *p* < 0.01), and burnout (r = 0.21; *p* < 0.001). Moreover, the results of this analysis demonstrated a positive association between envy and stress (r = 0.19; *p* < 0.001).

### 4.2. Structure Model Assessment

The model presented in this study was appropriate because all fitness indices were within reasonable bounds (see Table 3 and Figure 2). Various fitness indices were used to evaluate the model’s fit, including the Root Mean Square Error of Approximation (RMSEA), Tucker–Lewis Index (TLI), Comparative Fit Index (CFI), and Chi-square normalized by degrees of freedom. The proposed structural model showed a reasonable fit to the data with the following values: χ^2^ = 584.30, χ^2^/*df* = 1.63, CFI = 0.94, TLI = 0.93, RMSEA = 0.07 [81]. Table 4 presents the results of the standardized path coefficients for all hypothesized relationships between the variables.

Moreover, WO had an indirect positive effect (*β* = 0.03, 0.06, *p* < 0.01) on negative health outcomes (job stress and burnout) mediated by envy, with an acceptable range of 95% CI. Since this CI did not contain 0, we concluded that envy significantly mediated the relationship between WO and negative health outcomes (job stress and burnout). The direct positive effect of WO on burnout was 0.14 (*p* < 0.01); therefore, the mediating effect of envy was partial. The total effect of WO on negative health outcomes, that is, job stress and burnout, was 0.10 and 0.19, respectively.

## 5. Discussion

### 5.1. Theoretical Implications

Our study showed that employees who recognized WO within their organization were affected by job stress and burnout. Furthermore, employees’ self-protective perceptions of being the object of envy had a mediating effect on their WO. To achieve our research purpose, we conducted empirical research with 403 people in various companies. The results are summarized below.

First, it has been demonstrated that WO has a negative effect on health [46], job stress [82], and burnout among employees [83]. In line with the results of previous studies, we reconfirmed that WO is positively related to job stress and burnout in terms of employees’ health. As suggested by Sharma and Dhar [84], the fact that more research is needed to uncover the dangers of ostracism and the negative influence of WO, which needs to be managed more paradoxically, was also meaningfully reconfirmed by our findings.

Previous research has confirmed that job stress or burnout mediates the effect of WO on employees’ behavior and attitudes in various organizations. This study went a step further by revealing the mediating effect of perceiving envy to defend oneself. WO is a threatening situation for employees that can trigger the self-protective perception that they are the object of envy. Therefore, our study has important theoretical significance because it is the first to demonstrate the mediating effect of envy among employees in the Republic of Korea, where WO causes many social problems.

### 5.2. Practical Implications

In addition to its theoretical significance, this study offers the following practical implications. The study’s results confirmed that the perception of being the object of others’ envy was mediated to maintain a positive self-image in WO. However, despite this process, WO has a detrimental effect on employees’ health in an organization. It is important for leaders at the organizational level to recognize WO as a factor that needs to be managed and seek various action plans to this end. The following actions are suggested.

First, the health and psychological state of employees at the organizational level must not be neglected. Outcome variables, such as job stress and burnout, have been shown to affect performance through various previous studies. Therefore, checking employees’ health and psychological status in the organization on an annual or quarterly basis and ascertaining the frequency of experiences similar to WO is the most reliable way to confirm the impact of WO, which is not easily revealed.

Second, if WO is managed within the organization in terms of human resource development, cooperative and mutually beneficial behaviors among employees can be promoted by conducting group discussions or training that boost these behaviors. As part of this process, it is recommended to conduct training on how to use appropriate body language and multiple perspectives on communication [3]. In developing an organizational training program, a plan that can eliminate negative emotions or relationships should be sought. Through the implementation of such a program, employees can assist each other to cooperate constructively.

Those who instigate WO must be held accountable irrespective of their hierarchical reputation or unique talent. Additionally, providing stress relief options, such as a company fitness center, human resources hotline, or a conflict mediator, can encourage employees to develop a means to vent their pent-up emotions without passing them on to other employees. WO seems to be spreading due to competition between individuals as competition between companies intensifies due to technological advancement. This quiet but aggressive action requires more academic focus. We hope this study will contribute to sustainable organizations by triggering awareness of the negative interactions caused by WO.

## 6. Limitations and Strengths

As with all research, some limitations in our study should be considered. First, it is important to confirm the mediating effect of the recognition of envy in the impact of WO on health-related job stress and burnout among employees. However, attempting to maintain a positive self-image in a difficult environment within an organization can be an effort to survive. In this study, only the negative effects of WO were examined; however, as Balliet and Ferris [10] asserted, ostracism may also have the positive effect of promoting prosocial behaviors, for example, organizational citizenship behaviors, depending on the situation within the organization. Therefore, future research needs to confirm whether WO leads to both negative effects and altruistic behaviors, such as organizational citizenship behaviors, that are beneficial to organizations. Efforts in this direction will enable us to undertake new evaluations of the perception of envy that occurs in threatening situations within organizations. Second, in designing this WO study, a common method bias error was implied by using a questionnaire from the same respondent for all variables. To counter this limitation, a cross-sectional design was adopted, using the data collected by setting two different time points for questionnaire collection. Nevertheless, more sophisticated results could be obtained if the dependent variable is measured using more objective health-related third-party data. Third, this study targeted only office workers in Korea, and our findings should be verified in other countries.

## Figures and Tables

**Figure 1 behavsci-13-00669-f001:**
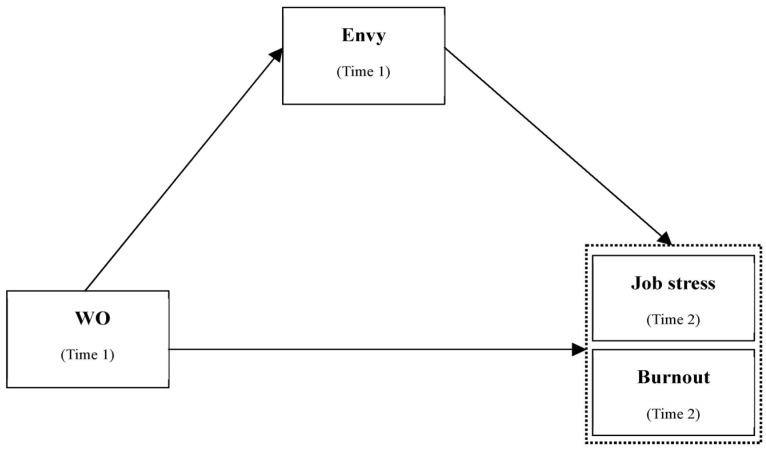
This study’s theoretical model. Notes: The two variables (job stress and burnout) inside the dotted line represent Negative Health Outcomes.

**Figure 2 behavsci-13-00669-f002:**
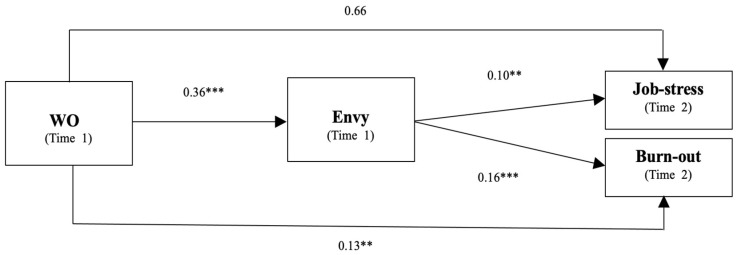
The research model with path coefficients. Note: ** *p* < 0.01, *** *p* < 0.001.

**Table 1 behavsci-13-00669-t001:** Descriptive features of the sample (*n* = 403).

Characteristics	Count	Percent
Gender		
Female	201	49.9%
Male	202	50.1%
Age (year)		
20–29	82	20.3%
30–39	175	43.4%
40–49	103	25.6%
50–59	43	10.7%
Job level		
Staff~Assistant	238	59.1%
Manager or deputy general manager	68	16.9%
Department manager	43	10.7%
Executive	54	13.4%
Tenure (years)		
1–4	40	9.9%
5–9	199	49.4%
10–14	84	20.8%
Over 15	80	19.9%

**Table 2 behavsci-13-00669-t002:** Means, standard deviations, correlations, and consistency coefficients for each variable.

	Mean	SD	1	2	3	4	α	AVE	CR
1. WO	1.99	1.00	1				0.95	0.82	0.95
2. Envy	2.80	0.96	0.33 ***	1			0.93	0.59	0.91
3. Job stress	3.03	0.84	0.17 **	0.19 ***	1		0.87	0.67	0.89
4. Burnout	3.07	0.89	0.21 ***	0.22 ***	0.71 ***	1	0.91	0.64	0.90

Notes: *n* = 403, list-wise deletion. Gender: male = 0, female = 1, WO = Workplace Ostracism. α = Cronbach’s alpha, AVE = average variance extracted, CR = composite reliability. ** *p* < 0.01, *** *p* < 0.001, two-tailed tests.

**Table 3 behavsci-13-00669-t003:** Results of the hypothesized model.

	Hypothesized Paths	β	S.E	t-Value
H1(a, b)	WO → Job stress	0.66	0.03	1.79
	WO → Burnout	0.13	0.04	2.99 **
H2	WO → Envy	0.36	0.05	6.77 ***
	Envy → Job stress	0.10	0.03	2.73 **
	Envy → Burnout	0.16	0.04	3.47 ***

Notes: *n* = 403, ** *p* < 0.01, *** *p* < 0.001.

**Table 4 behavsci-13-00669-t004:** Results of bootstrapped indirect effect tests.

	Direct Effect			Indirect Effect			Total Effect		
	Envy	Job Stress	Burnout	Envy	Job Stress	Burnout	Envy	Job Stress	Burnout
WO	0.36 **	0.07	0.14 **	-	0.03 *	0.06 **	0.36 **	0.10 **	0.19 **
Job stress	-	0.09 *	0.16 **	-	-	-	-	0.09 *	0.16 **

Notes: 10,000 times bootstrapped results are presented; * *p* < 0.1, ** *p* < 0.01, SE = standard error; BC = bias-corrected percentile method; CI = 95% confidence interval.

## Data Availability

Data supporting reported results are available from the authors on request.

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
