# Peer review of "Workplace Ostracism Effects on Employees’ Negative Health Outcomes: Focusing on the Mediating Role of Envy"

_behavsci, 2023, doi:10.3390/bs13080669_

Round 1
Reviewer 1 Report
This article examines the mediating effect of envy on workplace ostracism (WO) and employee health, providing a new perspective for employees facing workplace bullying. The study confirms the negative impact of WO on employee health, making it a valuable contribution to the field. At the same time, I have some minor suggestions for the improvement of the article.
In the abstract and introduction, the relationship between envy and WO needs further elaboration, as envy can exacerbate WO and affect employee physical health.
Additionally, on page 5, the study should consider that envy can also be a protective mechanism, according to STTUC.
The study collected data at two different time points using random sampling, but only selected 403 out of 950 participant responses for the final study. The number of participants in each stage of the study should be explained, and the reason for using only 403 responses for the final analysis should be provided. The large dropout rate could also be noted as a limitation.
Furthermore, due to the nature of study design, the study is limited to a correlation between variables, making it difficult to determine the direction of the relationship between envy, WO, and employee health. These issues could be considered as limitations.
Overall, this study is contributive. Further exploration of the cultural relevance of envy and its impact on employee health would be a worthwhile pursuit.
Author Response
Thank you for providing an opportunity for the thesis to be enhanced further based on your valuable advice.
I am attaching a response letter and manuscript.

Reviewer 2 Report
The study is interesting and certainly contributes to the development of psychology. However, a theoretical review is a bit one-sided focusing only on the negative effects of ostracism on employees whereas the phenomenon of ostracism is multifaceted.
Ostracism does not always lead to the intended effect. Some workers subject to these sanctions tend to violate the norm of social responsibility even more.
In open social communities, the effectiveness of ostracism goes to zero since the subject faces practically no obstacles to mobility between communities and previously formed social capital and reputation do not play a significant role in the process of transition.
At the micro level in the present-day socio-economic system, ostracism is virtually inapplicable due to the high vertical, horizontal, and territorial mobility of the population.
Ostracism is successful within professional communities where the support of their members is a key factor of an individual and professional success. Therefore, a threat of being excluded from the community forces subjects to comply with the established rules and to take into account not only personal interests but also the interests of other member of the community in the decision- making process.
One more important factor in the growth of the effectiveness of ostracism as a sanction is the subjects’ preferences for the future.
Within this topic, it seems strange not to see references to these publications:
Balliet D., Ferris D. L. (2013). Ostracism and Prosocial Behavior: A Social Dilemma Perspective. Organizational Behavior and Human Decision Processes, 120(2), 298–308. https://doi.org/10.1016/j.obhdp.2012.04.004
Balliet D., Parks C., Joireman J. (2009). Social Value Orientation and Cooperation in Social Dilemmas: A Meta-Analysis. Group Processes and Intergroup Relations, 12(4), 533–547. https://doi.org/10.1177/1368430209105040
and other works by Daniel Balliet.
The manuscript does not consider that ostracism may lead to the reinforcement of social values and preferences if actors have long-term expectations about the future, including a reasonable assessment of the prospects for interaction with community members. The subjects with a short planning horizon, on the contrary, consider ostracism to be an unfair punishment, reduce the degree of social interaction, and reinforce social dilemmas.
The empirical study
It is necessary to specify the employees of which companies participated in the study, how big these companies are, what organizational specificities they have.
What statuses do these employees have? What is their age? Gender?
As a result, apart from general conclusions, the authors were not able to obtain specific results on the impact of ostracism on people with different working experience, status, gender, peculiarities of corporate interaction, etc.
Author Response

(The authors gave the same response as above.)

Round 2
Reviewer 2 Report
The reviewer would like to thank the authors for carefully considering the comments and recommendations and is quite satisfied with the revisions made.
Author Response
We appreciate your review of our manuscript(behavsci-2472416).
Thank you for your valuable comments and recommendations which were extremely useful.
We revised minor reviews and censored typos.
I look forward to collaborating with you and the reviewers to ensure that the manuscript is acceptable for publication in Behavioral Sciences.
Yours sincerely,
